# The Utility of Fine Needle Aspiration (FNA) Biopsy in the Diagnosis of Mediastinal Lesions

**DOI:** 10.3390/diagnostics13142400

**Published:** 2023-07-18

**Authors:** Uma Kundu, Qiong Gan, Deepak Donthi, Nour Sneige

**Affiliations:** Section of Cytopathology, Department of Pathology, The University of Texas MD Anderson Cancer Center, Houston, TX 77030, USA

**Keywords:** FNA, mediastinum, cytology, mediastinal FNA

## Abstract

Fine needle aspiration is a minimally invasive, low-morbidity, and cost-efficient technique for the sampling of mediastinal lesions. Additionally, ancillary testing on FNA samples can be used for the refinement of diagnoses and for treatment-related purposes (flow cytometry, cytogenetics, immunohistochemistry, and molecular diagnostics). Mediastinal lesions, however, can show a variety of lineages and morphologic features, giving rise to diagnostic dilemmas. As a result, the differential diagnosis can vary widely and becomes especially challenging due to the smaller sample size on FNA and the variability in component sampling. For appropriate patient management and to determine the correct treatment strategies, accurate pathologic diagnoses are paramount. In this review, we present the cytomorphologic features together with the immunophenotypic findings of mediastinal lesions, with emphasis on the diagnostic challenges and pitfalls in FNA cytology samples, including smears and cell block sections.

## 1. Introduction

The mediastinum is considered the visceral compartment within the central thoracic cavity that extends from the diaphragm inferiorly to the thoracic inlet superiorly and contains vital organs and structures. A wide range of lesions may develop in the varying tissue components and organs in the mediastinum, prompting sampling in patients.

Although mediastinoscopy has traditionally been the gold standard for sampling mediastinal masses and lesions, current advances in imaging and sampling techniques have pushed computed tomography (CT)-guided fine-needle aspiration (FNA) and endoscopic ultrasound-guided (EUS) FNA to the forefront of sampling these masses and lesions. FNA is minimally invasive, cost-efficient, and associated with low morbidity. Mediastinal lesions can show a variety of lineages and morphologic features, causing diagnostic dilemmas. For example, solid lesions such as thymic epithelial lesions and neoplasms, lymphomas, and germ cell tumors have overlapping features and/or biphasic patterns that encompass neoplastic and even non-neoplastic components. As a result, differential diagnoses can vary widely and are especially challenging due to the smaller size of samples collected by FNA, variability in component sampling, and lack of histologic architecture. Additionally, ancillary testing on FNA samples can be used to refine diagnoses and to make treatment-related decisions (e.g., flow cytometry, cytogenetics, immunohistochemistry, and molecular diagnostics). To determine appropriate treatment strategies, accurate pathologic diagnoses are paramount.

In this review, we present the cytomorphologic features and immunophenotypic findings of mediastinal lesions with an emphasis on the diagnostic challenges and pitfalls of FNA cytology samples.

### 1.1. Specimen Procurement and Triage

FNA samples are usually procured by a radiologist using image guidance (CT or endoscopic ultrasound). Using a 20-gauge or 22-gauge needle, one to two needle passes per case are performed. One part of the specimen is smeared directly on glass slides, and the remaining material is rinsed in Roswell Park Memorial Institute medium or a cytolyte solution. The direct smears are air-dried for Diff-Quik staining (Stat Lab, Lewisville, TX, USA) or fixed in alcohol or modified Carnoy fixative (in a 6:1 ratio of 70% ethanol to glacial acetic acid) for rapid Papanicolaou staining. Rapid on-site evaluation (ROSE) is performed by the cytopathologist in most cases. The needle rinse is evaluated and processed either for cytospin, ThinPrep, or cell block preparation, depending on the material present.

Should a lymphoproliferative disorder have been suspected after ROSE, a portion of the FNA specimen is collected in Roswell Park Memorial Insitute-1640 medium with 1% fetal bovine serum, and a cell count is performed and submitted for immunophenotyping by flow cytometry. Portions of the FNA specimen are also saved for possible molecular or cytogenetic testing in relevant cases.

### 1.2. Cytologic Characteristics

For the purpose of this review, FNA specimens can be classified as 1. non-diagnostic; 2. non-neoplastic, which includes cysts, inflammation, ectopic tissue, and thymic hyperplasia; and 3. neoplastic, which is further divided into A. epithelial neoplasms such as A1. thymic epithelial neoplasms (thymomas and thymic carcinomas), A2. neuroendocrine neoplasms and A3 metastatic carcinomas, B. mesenchymal neoplasms (B1. Neural, B2. Smooth muscle B3. Lipomatous B4. Vascular and B5. Other spindle cell neoplasms), C. germ cell tumors (C1. seminoma and C2. non-seminomatous tumors), and D. hematologic malignancies (Figure 1). Non-diagnostic specimens are defined as non-lesion tissues, cyst contents with histiocytes discordant with imaging findings, and/or specimens with obscuring blood or distortion artifacts. This review will focus on non-neoplastic and neoplastic specimens.

## 2. Non-Neoplastic Lesions

### 2.1. Cystic Lesions

Cystic lesions of the mediastinum make up between 15% and 30% of all mediastinal masses, with most representing benign entities [1]. Patients with mediastinal cysts are generally asymptomatic, and these lesions can be found incidentally on CT imaging and mistaken for a solid mass [2]. The diagnosis of a benign cyst via EUS-FNA may prevent unnecessary thoracic surgery in asymptomatic patients. However, some patients may appear symptomatic owing to compression of adjacent vital structures or secondary infection [3]. EUS-FNA can decompress cysts, and surgical resection is considered curative [4]. The development of primary mediastinal cysts is largely embryologically related. For example, foregut cysts, such as bronchogenic cysts, arise from a developmental abnormality of the tracheobronchial tract and frequently occur in the middle mediastinal compartment. “Fluid-thrill sign” on EUS has been recently described for the recognition of bronchogenic cysts when CT findings are similar to those of a soft tissue mass [5].

Foregut cysts also include enteric cysts (i.e., esophageal duplication cysts) that tend to arise from an abnormality in development during the fusion of the tracheoesophageal septum. These cysts occur more commonly in the posterior mediastinal compartment. EUS can delineate the multiple layers of duplication cysts that have been ascribed histologically to double layers of smooth muscle [6,7].

The cytology of aspirated cysts includes the recognition of macrophages, proteinaceous fluid, degenerated cells, amorphous debris, and possibly hemosiderin. Identifying the specific cysts that line the epithelium can aid in the diagnosis of certain cysts, such as bronchogenic cysts, which contain ciliated columnar epithelium or bronchial-type cells (Figure 2). Bronchogenic cysts may also contain seromucinous glands, fragments of islands of cartilage, or fascicles of smooth muscle, which can cause diagnostic confusion with normal tracheobronchial structures when sampled using EBUS-FNA [6,8]. Recognizing background cystic changes and correlating the changes to radiologic findings can be helpful in diagnosing cysts. Duplication cysts can show ciliary tufts detached from the epithelial cells or stratified squamous epithelium, and seromucinous fluid may or may not be present [7,9].

Ciliated columnar cells can be seen in the recently described mediastinal Mullerian cyst. These cells can often be confused with bronchogenic cysts, and it is possible that they were historically misclassified as such. Location within the mediastinum provides guidance for an appropriate diagnosis, as most bronchogenic cysts occur in the middle mediastinum, while Mullerian cysts tend to occur in the posterior mediastinum. Additionally, should a cell block be available for immunohistochemical staining, the cells of the epithelial lining are reported to be positive for PAX-8, WT-1, ER, and PR [10,11].

Thymic cysts can arise from a variety of processes in the anterior mediastinum, including congenital, inflammatory, or even neoplastic processes, that can present in unilocular or multilocular forms. Congenital thymic cysts arise from residual tissues of the third pharyngeal pouch that persist as a unilocular lesion, while acquired cysts from inflammatory processes tend to be multilocular [8,12,13]. FNA of thymic cysts demonstrates predominantly cyst contents and may be considered non-diagnostic unless epithelium-lining tissues and/or thymic tissues are also present. A major pitfall in diagnosing cystic lesions includes cystic changes within a neoplastic process, such as cystic thymomas, and therefore correlation with radiologic results can be utilized (Figure 2).

### 2.2. Inflammatory Lesions

#### Mediastinitis and Granulomatous Inflammation

ROSE is particularly helpful when microbial cultures are required because samples can be submitted immediately for cultures. Acute mediastinitis can occur as iatrogenic perforation or as descending necrotizing mediastinitis from the oropharyngeal flora. Fibrinopurulent fragments, bacterial colonies, and necrosis may be present in mediastinitis [14]. Aggregates of epithelioid to spindled histiocytes with mixed background inflammatory cells, including eosinophils and lymphocytes, can be seen in cases of chronic and/or granulomatous inflammation (Figure 3). Areas of necrosis or caseation may be present. Special stains for acid-fast and fungal organisms can be performed on cell block sections to aid in the diagnosis of an infection. “Clean granulomas”, in which background inflammatory debris is not identified, may be indicative of sarcoidosis [15,16]. In either case, correlating these findings with patient history is warranted. Chronic/sclerosing mediastinitis often leads to non-diagnostic or acellular smears due to the fibrosis present in these lesions or to non-specific findings, such as fibroinflammatory fragments, dense collagen fragments, or calcifications [3,17]. Sclerosing mediastinitis is thought to arise from various causes, including autoimmune disorders, such as rheumatoid arthritis; inflammatory disorders; infectious processes, such as histoplasma; and malignancies (Figure 3).

### 2.3. Ectopic Tissue

FNA in mediastinal lesions can lead to the diagnosis of benign tissue in unexpected locations. As the thyroid descends during embryological development, aberrations in migration may result in remnants of thyroid tissue within the mediastinum. Although rare, mediastinal ectopic thyroid tissue can occur in 1% of mediastinal masses and usually occur within the anterior mediastinum [18]. In these tissues, follicular epithelial cells with round, smooth nuclei and a background colloid are present (Figure 4). Care should be taken to note their nuclear features, as the oval-shaped nuclei, nuclear grooves, and intranuclear inclusions would raise concerns for papillary thyroid carcinoma. Other tumors, such as carcinoid tumors, can have a morphologically monotonous appearance akin to follicular epithelial cells. Carcinoid tumor cells on FNA smears are larger than follicular epithelial cells and demonstrate a salt-and-pepper chromatin pattern with a lack of background colloid [18]. Similarly, the aberrant findings of parathyroid tissue within the mediastinum may be related to the embryological descent of the thymus. In cases of parathyroid adenomas of the mediastinum, patients may present with hypercalcemia or elevated parathormone levels [19,20]. Cytologic features include loose clusters of cells or even naked nuclei with well-defined cell borders. The cells can be polygonal with an oxyphilic cytoplasm. Their chromatin pattern is stippled, and anisokaryosis can be present along with eccentric nuclei [19,20,21]. As fibrovascular cores can occasionally be seen with parathyroid adenomas, diagnostic dilemmas can occur in patients with a history of papillary thyroid carcinoma. Closer inspection for anisokaryosis and determining the chromatin pattern can be helpful, along with considering any clinical history of hypercalcemia or hyperparathyroidism.

### 2.4. Thymic Hyperplasia

The maturation of T-lymphocytes occurs within the thymus, which is a specialized lymphoepithelial organ located in the anterior mediastinum. Thymic hyperplasia can occur as true thymic hyperplasia (TTH) or thymic lymphoid hyperplasia (TLH). Thymic hyperplasia primarily occurs in children, and if the patient is less than 3 years old, they may present with an enlarged thymus gland, which can resolve on its own [22]. In children who have received chemotherapy for malignancies or are recovering from illness, thymic hyperplasia presents as an anterior mediastinal mass and may cause alarm among clinical teams that the child’s malignancy has recurred [23,24]. As a minimally invasive procedure, FNA can distinguish thymic hyperplasia from malignancy, preventing the use of invasive procedures or treatments [25]. Smears of TTH have lymphocytes in various stages of maturation and a mixture of epithelial cells, or Hassall corpuscles, that are positive for pan-cytokeratin, p63, and p40 [25,26]. Unlike in thymoma, the lymphocytes in thymic hyperplasia tend to encircle individual epithelial cells [27]. TLH occurs in children, young adults, and patients with autoimmune disorders such as myasthenia gravis [28,29,30]. Cytology smears of TLH contain lymphocytes in various stages of maturation, along with tingible body macrophages and germinal centers [26]. As thymomas generally occur in adults, the patient’s age along with correlating this information with that of imaging studies and clinical data, can be used to distinguish between thymomas and thymic hyperplasia.

## 3. Neoplastic Lesions

Neoplastic lesions of the mediastinum can be classified as epithelial Section 3.1. epithelial, Section 3.2 mesenchymal; Section 3.3 germ cell tumors; and Section 3.4 hematologic. Table 1 provides a full list of common abbreviations for mediastinal neoplasms.

### 3.1. Epithelial Neoplasms

Epithelial neoplasms can be further classified as Section 3.1.1 primary thymic epithelial neoplasms, including thymomas and thymic carcinomas, and Section 3.1.2 neuroendocrine neoplasms.

#### 3.1.1. Thymic Epithelial Neoplasms

Thymic epithelial neoplasms (including thymomas and thymic carcinomas) are the most common anterior mediastinal masses and occur mainly in adults. The vast majority of thymomas are asymptomatic; however, larger tumors can have compressive symptoms. Paraneoplastic syndromes are also associated with thymomas such as myasthenia gravis. Thymic epithelial neoplasms (TEN) have a component of neoplastic epithelial cells and a component of predominantly small lymphoid cells with mature and immature T-cell phenotypes. The diagnostic challenges of using FNA arise from the variable proportions of the dual components, which can be intrinsic to the tumor or caused by sampling errors. Other challenges arise related to the heterogeneity of their morphologic appearances and the lack of spatial orientation on FNA samples compared with that of histology.

The updated 2021 WHO classification for thymic tumors delineates several categories and molecular advancements, including not otherwise specified thymoma; thymoma types A, AB, B1, B2, and B3; micronodular thymoma with lymphoid stroma; metaplastic thymoma; and lipofibroadenoma. Type A and Type AB thymomas show a high frequency of GTF2I (p.L424H) mutations, whereas loss of function mutations in TP53 are more typical of type B thymomas and thymic carcinomas [31]. Our institution uses the Suster-Moran classification to classify thymic epithelial tumors as thymomas, atypical thymomas, or thymic carcinomas, which is based on the presence or absence of organotypic features of differentiation along with epithelial cell atypia [32,33].

Aspirates of TEN can be moderately to highly cellular. The neoplastic epithelial component may vary from loosely spindled to ovoid cells with homogenous chromatin and indistinct nucleoli, at times forming whirling arrangements in spindle cell thymomas, to sheets or loose epithelial cells with round to oval nuclei in a background of lymphocytes. Depending on the most common cell type, the differential diagnosis may change. For example, spindle cell thymomas raise the possibility of mesenchymal neoplasms, such as smooth muscle tumors, solitary fibrous tumors, and neuroendocrine tumors, with spindle cell morphology [34]. In addition, fibrous tissue or sclerosis within the mediastinum can cause further diagnostic dilemmas. A panel of immunohistochemical studies can be helpful for cell block preparations.

In lymphocyte-predominant thymomas, recognition of epithelial groups may be difficult because they can occur as isolated cells or tightly clustered microfragments [34,35]. The clustering and tight overlap of epithelial cells can be challenging to recognize, and in these cases, cell block preparation with immunohistochemical staining may be employed (Figure 5). The epithelial fragments are positive for pan-cytokeratin, CK5/6, p40, and p63, while the small reactive lymphocytes (i.e., immature T-lymphocytes) are positive for terminal deoxynucleotidyl transferase, CD1a, and CD3. On the one hand, CD5 can be expressed in thymoma lymphocytes, but on the other hand, it has staining in the epithelial cells of thymic carcinomas [34,36]. The preponderance of lymphoid tissue raises the differential diagnosis of reactive lymphoid hyperplasia or lymphoma. Polymorphous lymphocytes with tangible-body macrophages and follicular center cell fragments can be seen in reactive lymphoid hyperplasia without an epithelial component. However, germinal center fragments and macrophages may be confused for an epithelial component. Therefore, recognizing a second population of epithelial cells is necessary, ideally with immunohistochemical confirmation on cell block preparations or de-stained Papanicolaou smears [35]. ROSE can be particularly helpful in triaging cases when working up lymphoma, including submitting samples for flow cytometric analysis. Notably, immature thymocytes are similar to T-lymphoblastic leukemia/lymphoma (T-LL) blasts on cytology smears, further causing a diagnostic conundrum. Thymocytes usually exhibit a spectrum of maturation with immature and mature-appearing lymphocytes, which is reflected in flow cytometric analyses of a gradation pattern of surface CD3, CD45, CD4, and CD8 [33,37] (Figure 6). In contrast, T-LLs are more monotonous, are larger with irregular nuclear contours, have evenly distributed chromatin, and show a cluster pattern of CD3, CD45, CD4, and CD8 on flow cytometry [33,37]. Notably, both entities can show expression of CD1a; cytoplasmic CD3; partial surface CD3, TdT, CD34, CD10, CD45 (dim), and CD99; and double expression of CD4/CD8. Aberrant expression of CD10 and CD34 can be seen in T-LLs along with a loss of expression of T-cell antigens, which is not commonly seen in thymocytes. [33,37,38]. Clinical history can also provide guidance, as T-LL is more commonly seen in younger patients and thymoma is rare.

Cases of thymoma in which both lymphoid and neoplastic epithelial components are equally present are easier to recognize owing to their geographic fragments of tissue and endothelial wrapping of the fragments (Figure 7). However, entities such as mediastinal seminomas can also present with a dual population of neoplastic cells against a background of lymphocytes, which is similar to the presentation of thymoma. The seminoma cells cling in aggregate to branching capillaries and are generally larger and more pleomorphic/polygonal with an abundant amount of cytoplasm, round nuclei, and multiple prominent nucleoli compared with the characteristics of epithelial thymoma cells [39]. The hallmark tigroid background or epithelioid histiocytes are not always seen on FNA smears [40,41]. Additionally, Hodgkin lymphomas appear to have a dual population in sparse FNA aspirates. A careful search for Reed-Sternberg cells, along with recognizing the lymphoid background as polymorphous with eosinophils and plasma cells, is paramount in differentiating Hodgkin lymphoma and thymomas. Epithelial-predominant thymomas frequently have distinct cell borders, round to oval nuclei, and cytoplasm with a squamoid appearance. The epithelial fragments may appear “geographic” or similar to a “jig saw puzzle piece” [26]. A careful search for possible background lymphocytes is helpful, but they may be few and far between. Compared with neuroendocrine tumors such as carcinoid tumors, which can also have a bland appearance, the nuclear chromatin of the epithelial cells of thymoma is not of a salt-and-pepper chromatin pattern, and the cytoplasm is less granular (Figure 8). Thymomas presenting with a pseudorosetting pattern on FNA have also been reported and can be a mimicker of neuroendocrine tumors [34].

Thymic carcinomas on FNAs exhibit features similar to their malignant counterparts derived from other organs and vary cytologically by subtype. For example, squamous cell carcinoma, the most commonly reported thymic carcinoma in the literature, shows clearly malignant cells with a coarse chromatin pattern and a moderate amount of cytoplasm with or without keratinization [42]. As mentioned previously, thymic carcinomas are positive for CD5 and CD117, whereas these immunohistochemical stains on cell block preparations are negative in thymomas [34]. When distinguishing thymic carcinomas from lung primary squamous carcinomas, determining PAX8, CD5, and CD117 positivity in thymic carcinomas can be useful to distinguish thymic carcinomas from metastatic carcinomas [43].

Additional subtypes include adenocarcinomas, clear cell carcinomas, basaloid carcinomas, lymphoepithelial-like carcinomas, and sarcomatoid carcinomas. *NUT* carcinomas (Figure 9) are particularly aggressive carcinomas of the mediastinum and portend poor clinical outcomes. These carcinomas demonstrate genetic rearrangements of *NUT*. Chromosomal translocation between *NUTM1* on chromosome 15q14 and *BRD4* on chromosome 19p13.1 are more common than those with *BRD3* on chromosome 9q34.2, among other fusion partners that have been detected. Procurement of material for ancillary testing during ROSE is important as the translocations are detectable by fluorescence in situ hybridization (FISH). This procurement can include cell block material for *NUT* immunohistochemistry or for gene fusion detection as well. FNA smears for NUT carcinoma are usually cellular, with round cells that appear discohesive, or “primitive”, with variation in cytoplasmic qualities from delicate to dense. Pleomorphism and multinucleation can be present along with focal keratinization. Nuclei are minimally irregular, and background necrosis and karryorhetic debris are usually evident [44,45].

#### 3.1.2. Neuroendocrine Neoplasms 

Primary neuroendocrine neoplasms of the mediastinum, albeit rare, traditionally require a more aggressive clinical course than their pulmonary counterparts. Association with multiple endocrine neoplasia syndrome (MENS) has been noted in the literature along with other paraneoplastic syndromes, such as Cushing Syndrome and Eaton-Lambert Syndrome [46]. Association with MENS or an endocrinopathy has been shown to be a poor prognostic factor, with a 5-year survival rate of 35% compared with that without endocrinopathy, which is 65% [47,48,49]. Interestingly, these neoplasms are not associated with carcinoid syndrome.

Cytologically, these tumors can be further divided into well-differentiated neuroendocrine carcinoma, or carcinoid tumors; moderately differentiated neuroendocrine carcinoma, or atypical carcinoid tumors; and poorly differentiated neuroendocrine carcinomas, which include small cell and large cell neuroendocrine carcinoma [50].

Cytology smears of well-differentiated neuroendocrine carcinoma are characterized by clusters of uniform round-to-ovoid cells in a pseudorosette pattern [51,52]. Single cells with scant cytoplasm or even stripped nuclei may be present. A stippled, or a “salt-and-pepper”, chromatin pattern is usually evident cytologically, along with a lack of background necrosis (Figure 10). Neuroendocrine carcinomas are positive for INSM1, synaptophysin, and chromogranin, while the epithelial component in TEN is positive for CK5/6, p63, and p40. Hou and colleagues have demonstrated that INSM1 is a reliable neuroendocrine marker in cytology smears, with an overall detection rate of 94% [53]. Some neuroendocrine tumors can also demonstrate a spindle cell pattern, which can pose challenges when differentiating spindle cell thymomas from mesenchymal tumors. Additionally, close clinical and radiological correlation is needed to exclude the possibility of a pulmonary primary.

Moderately differentiated neuroendocrine carcinomas show more pleomorphism or cellular atypia, a salt-and-pepper or dusty chromatin pattern, and mitoses (Figure 11). The high nucleus-to-cytoplasm ratio for small-cell carcinomas is notable on cytology, along with a smear effect. Small cell carcinomas demonstrate nuclear molding, a stippled chromatin pattern, a higher mitotic rate, and areas of necrosis (Figure 11). The cells of large-cell neuroendocrine carcinoma contain a moderate amount of cytoplasm, round-to-oval nuclei with prominent nucleoli, increased mitotic rates, and areas of necrosis. The abundant cytoplasm and prominent nucleoli can bring adenocarcinomas into the differential diagnosis.

### 3.2. Mesenchymal Neoplasms

Mesenchymal neoplasms of the mediastinum are generally considered rare and arise from various structures or derivations, such as neural structures and fibroadipose tissue. A series by Abdel-Rahman confirmed not otherwise specified sarcoma as the most common category for primary mediastinal sarcomas [54]. Such categorizations may be due to limited sampling or aspects inherent to the lesion itself after ancillary testing, such as FISH, immunohistochemical studies, and molecular diagnostics, fails to further categorize these lesions. Patients with primary mediastinal sarcomas have a worse overall 10-year survival rate compared with non-mediastinal sarcomas (23% vs. 55%, respectively) and compared with other mediastinal malignancies [54]. Mesenchymal neoplasms can be further characterized into the following categories:

#### 3.2.1. Neural Neoplasms

Neurogenic tumors may arise along peripheral nerves or nerve roots or anywhere along the sympathetic chain from sympathetic and parasympathetic ganglia. Most cases of neural neoplasms develop within the posterior mediastinum, with the most common entity being schwannoma [55,56]. Extension into the neural foramen gives the classic dumbbell appearance on imaging studies, such as CT or magnetic resonance imaging, complicating the approach for neural surgeons as complete resection is generally considered curative for peripheral nerve sheath tumors [57]. Although 90% of schwannomas occur in isolation, multiple schwannomas should raise concern for associated syndromes such as schwannomatosis, neurofibromatosis type 2, and the Carney complex [58]. FNA aspirates of schwannoma show cohesive clusters of spindled to plump cells in tissue fragments; isolated single cells are generally not encountered. The nuclei are spindled and pointed, at times wavy, and rarely show intranuclear cytoplasmic pseudo-inclusions or nuclear palisading (Figure 12). Although Verocay bodies are difficult to appreciate on cytology, procurement of material for cell block preparation is helpful not only for this histologic feature but also to perform immunostains that are known to be positive in schwannomas, such as S100 and SOX10. The cells are suspended within a fibrillary background that is best appreciated on Diff-Quik smears. Degenerative changes that can lead to nuclear atypia or “ancient schwannomas” should not be confused for a more ominous process [59].

Arising from neural crest cells of the sympathetic ganglia, especially within the posterior mediastinum, ganglioneuromas also possess “Schwannian stroma” composed of fragments of clustered spindled cells with wavy nuclei and indistinct cell borders (Figure 12). The key distinction is the presence of polygonal cells with granular cytoplasm, round, smooth, contoured nuclei, and prominent nucleoli that represent ganglion cells (Figure 12). Aggregates of lymphocytes may be present in ganglioneuromas, and therefore confusion with neuroblasts is cautioned on FNA samples to prevent misinterpretation as ganglioneuroblastoma [60]. The neuroblasts will have high nuclear-to-cytoplasmic ratios and a coarse chromatin pattern, and at times the nuclei may appear “smudged” [61]. The neuroblasts may form rosettes and show varying forms of maturation [62]. The patient’s age, is certainly helpful, as ganglioneuroblastomas and neuroblastomas occur in childhood, and although the majority arise within the adrenal gland, they also arise in the posterior mediastinum. Neuroblastomas represent the immature portion of the spectrum and are the fourth most common childhood tumor [63,64]. Patients may present with elevated urine catecholamines. Cytology smears show small to intermediate-sized cells, in loose clusters or singly, with very little distinction between cell borders and fibrillary cytoplasm. The cells can be arranged in Homer Wright pseudorosettes with increased mitoses and karyorrhectic debris. The nuclei are generally round with a salt-and-pepper chromatin pattern. Immunohistochemistry on cell block sections can show positive staining for synaptophysin and chromogranin; however, 20% of alveolar rhabdomyosarcomas and a subset of Ewing sarcomas can also stain positive. Recently, GATA3 has been shown to provide strong, diffuse staining in neuroblastoma while being negative in Ewing sarcoma, rhabdomyosarcoma, and myxoid round cell tumors [65]. NMYC and ALK amplification are associated with a poorer prognosis in sporadic neuroblastomas [66]. The mitotic-karyorrhectic index has been shown to have strong concordance between FNA smears and core biopsy specimens (86.7%) for classification into prognostic groups [64].

Another benign peripheral nerve sheath tumor, neurofibroma, develops often in patients with NF1 and classically arises within the posterior mediastinum (Figure 13). Neurofibromas are composed of Schwann cells with ill-defined cell borders that can be arranged in ropy fascicles with wavy nuclei, fibroblasts, and mast cells [67]. Pitfalls in fine needle aspirates of neurofibromas include that the smears can be non-diagnostic due to the tenacity of the lesion itself and low cellularity from hyalinized, myxoid, or central cystic/degenerated areas, similarities with schwannoma, or even sampling of lower-grade spindled areas of higher-grade lesions if all components are not represented [67].

Patients with NF1, particularly those who present with the plexiform subtype of neurofibroma, can have tumors that progress to overt malignant change (malignant peripheral nerve sheath tumor), which occurs in 8–15% of patients [68]. Mere cytologic atypia should be distinguished from malignancy. The spectrum of changes towards malignancy has been elucidated in a category of atypical neurofibroma of uncertain biologic potential by Miettinen and colleagues. The criteria not only consider increased cellularity and cytologic atypia but also loss of architecture and mitotic activity greater than 1 mitotic figure per 50 high power fields but less than 3 mitotic figures per 10 high power fields.

Rising along the sympathetic and parasympathetic chain paraganglia, which are derived from the neural crest cells, paragangliomas can be a rare cause of hypertension in children with elevated plasma free metanephrine levels; however, presentation in middle-aged adults is also common [69]. The cytology of paraganglioma shows single or clustered cells, some with an acinar-like arrangement, a few plasmacytoid cells, and round to oval nuclei with anisonucleosis. Rare intranuclear inclusions and/or nuclear pleomorphism are present, which may cause diagnostic confusion with thyroid neoplasms or melanoma [70]. Additionally, plasmacytoid features may also bring to mind medullary thyroid carcinoma [71]. The delicate cytoplasm and/or cytoplasmic vacuoles can also be confused for metastatic renal cell carcinoma [72]. Overall, paragangliomas are negative for cytokeratins such as AE1/AE3 and CAM5.2 while positive for GATA3, which can aid in distinction with an appropriate immunohistochemistry panel [73].

#### 3.2.2. Smooth Muscle Neoplasms

When considering spindle cell neoplasms besides neurogenic tumors, one can also consider smooth muscle tumors, especially tumors arising in the posterior mediastinum when imaging demonstrates no distinct relationship with nerves or nerve roots [74]. Smooth muscle tumors can also develop from the esophageal wall or great vessels within the mediastinum or, rarely, unconnected to any evident structure, perhaps due to misplaced mesoderm during embryonic development [74,75]. In truth, the majority of leiomyosarcomas in the mediastinum likely represent metastatic disease. As with all smooth muscle tumors elsewhere in the body, FNA smears show aggregates of spindle cells arranged in intersecting fascicular fashion in a dense eosinophilic matrix with “cigar-shaped” hyperchromatic nuclei (Figure 14). Cases of leiomyosarcoma demonstrate greater pleomorphism, cytologic atypia, increased mitotic activity, and areas of necrosis [76,77]. Paranuclear vacuoles may also be seen on cell block sections. Positive immunohistochemical staining on cell block sections for SMA and desmin is also helpful.

#### 3.2.3. Lipomatous Tumors

CT imaging of the mediastinum indicating a lobulated soft tissue mass or features of fat density, especially in a child or young adult, can point to the possibility of thymolipoma—particularly in the anterior mediastinum, although lipomatous tumors can occur across all mediastinal compartments [31,78,79]. Thymolipomas on FNA show lymphocytes and epithelial cells consistent with thymic tissue. Adipose tissue with abundant cytoplasm, different-sized vacuoles, and small regular nuclei are present; however, one cytology report illustrated the lack of adipose tissue in their case, and another report illustrated that mild atypia within the adipose tissue led to the erroneous diagnosis of liposarcoma on FNA [79,80]. However, should atypically large nuclei be observed within or next to vacuolated fat cells on FNA smears, the possibility of a well-differentiated liposarcoma cannot be excluded [81]. In these cases, cell block preparation with MDM2 and CDK4 immunohistochemistry can be used to assess for positive nuclear staining, along with FISH studies for amplification of MDM2 for WDLPS (negative in thymolipoma or lipoma) [31]. These genetic alterations are also seen in dedifferentiated liposarcoma [31]. Dedifferentiated liposarcomas demonstrate areas of transition to poorly differentiated sarcomas, and up to 10% can have heterologous differentiation [31,82]. Although WDLPS and DDLPS are more common than other subtypes, myxoid liposarcoma and pleomorphic liposarcoma can occur. Myxoid liposarcoma shows cellular smears with ovoid and spindle shaped cells, uniform nuclei, arborizing capillary networks, and a myxoid background. The lipoblasts are univacuolated with scalloped nuclei [83]. In one study, cell block preparations and unstained smears were successfully used for FISH analysis of DDIT3 rearrangements in 13 tested cases [84]. Last but not least, pleomorphic liposarcoma is seldom encountered in the mediastinum, with only a few scattered reports in the cytology literature. FNA smears of pleomorphic liposarcoma are described as cellular, with epithelioid to spindled cells, multinucleated tumor giant cells, and lipid-filled cytoplasmic vacuoles [85]. Although prominent pleomorphism favors PLPS, this feature can also be seen in DDLPS; therefore, the coco-expression of MDM2 and CDK4 is helpful to identify DDLPS [86].

#### 3.2.4. Vascular Tumors

Vascular tumors include hemangioma, epithelioid hemangioendothelioma, and angiosarcoma. FNA smears are usually hypocellular. Such findings are commensurate with the challenges expected in FNAs of vascular lesions, which result in bloody/hemorrhagic aspirates and few cells, and FNA detection rates of angiosarcomas have been reported at 53% [87,88]. Mediastinal hemangiomas are more common in patients younger than 35 years and can occur in the anterior or posterior mediastinal compartments [89]. On CT or magnetic resonance imaging, the presence of phleboliths, a pampiniform growth pattern, and aberrant draining veins are characteristic features of mediastinal hemangiomas [89,90]. Epithelioid hemangioendothelioma, considered a low- to intermediate-grade vascular malignancy, is composed of epithelioid to spindled cells with eosinophilic cytoplasm in clusters or individually, with some surrounding lumina that contain red blood cells. Nuclear grooves or intracytoplasmic vacuoles (i.e., blister cells) may also be present within a background of myxoid-like or extracellular matrix material (Figure 15). Cell block preparations with immunohistochemical vascular markers positive for CD31, CD34, and ERG are helpful. Some epithelioid hemangioendothelioma demonstrate a fusion of *WWTR1* and *CAMTA1CAMTA1*, and immunohistochemistry for *CAMTA1* can be positive, while others with *YAP1*-*TFE3* fusion show nuclear expression of TFE3 [91,92]. In one study, all angiosarcomas were negative for *CAMTA1*, allowing for the distinction between angiosarcomas and epithelioid hemangioendotheliomas, of which 90% of cases were positive for *CAMTA1* [93]. FNA smears of angiosarcoma can show epithelioid or spindled cells, which are incohesive at times and in single forms, and irregular pleomorphic nuclei with prominent nucleoli [94]. Spider legs or tadpole-like cytoplasmic extensions have been described in the literature [94,95]. Tumor giant cells can be present along with background necrosis and increased mitotic activity. Cytoplasmic vacuoles, or cells engulfing neutrophils and red blood cells, are characteristic of angiosarcoma hemangioendothelioma [96]. A lack of an extracellular matrix in angiosarcoma is also a helpful feature when distinguishing it from epithelioid hemangioendothelioma [97]. The differential diagnosis at this anatomic site includes mesothelioma and carcinoma; however, vasoformative channels should not be seen in these entities. Care should be taken in interpreting pan-cytokeratin, as up to 29% of epithelioid hemangioendotheliomas and 25% of angiosarcomas can be cytokeratin-positive, and therefore a comprehensive panel is suggested [98].

#### 3.2.5. Other Spindle Cell Neoplasms

Solitary fibrous tumors (SFTs) with their characteristic *NAB2*-*STAT6* fusion gene can present on FNA smears as irregular clusters of spindled to ovoid cells along collagenous-type stroma and staghorn-branching vasculature (Figure 16). Dissociated nuclei surrounding these aggregates have also been described [99,100,101]. Yet, we have observed that cytopathologists may not be able to ignore the fact that SFT is rather nondescript and cytomorphologically overlaps with many entities, and this view is supported by the literature. Although regarded as neoplasms of uncertain malignant potential, single cells on cytology may denote malignancy, while many cases of low malignant potential SFT also exhibit single cells on cytology [101,102]. Tani and colleagues were not able to correlate the degree of atypia with tumor metastasis. Cell block sections may meet the proposed criteria of malignancy of hypercellularity (>4 mitoses per 10 high powered fields, as defined by the World Health Organization), and the margins of necrosis—however infiltrative—cannot be assessed on FNA. STAT6 can be used as a surrogate marker for the fusion gene, with a reported sensitivity of >95% [103,104]. It is important to note that a diagnostic pitfall includes STAT6-positive nuclear staining in dedifferentiated liposarcomas and desmoid tumors and even weak nuclear staining in synovial sarcomas, which can also demonstrate spindle cell morphology on FNA smear [104].

Primary mediastinal synovial sarcomas are rare, accounting for 9–11% of intrathoracic synovial sarcomas, occur in all mediastinal compartments, and are predominantly monophasic rather than biphasic [105,106,107,108]. The monophasic form of synovial sarcomas shows a monotonous cellular proliferation of spindle to ovoid cells with scant cytoplasm and smooth elongated nuclei. Numerous background naked nuclei and increased mitotic activity can also be seen, although cytologic atypia is rather minimal (Figure 16). Thus, diagnostic dilemmas occur when differentiating possible sarcomatoid carcinomas and spindle cell thymomas; however, keratin expression in synovial sarcomas will not be diffuse. Furthermore, as discussed above, STAT6 immunohistochemistry can be employed to distinguish SFT, mitotic activity should be absent from schwannomas, and clinical history is paramount in the differential diagnosis of neurofibromas [109]. Additionally, synovial sarcomas demonstrate a FISH- or RT-PCR–detectable translocation of t(X;18)(p11;q11), leading to the fusion of *SS18* and *SSX1* and strong/diffuse TLE1 expression, which can be determined via immunohistochemistry [109,110]. The cytology of the biphasic form of synovial sarcoma shows an additional epithelioid component that can form glandular-like structures composed of round to cuboidal cells and an abundant cytoplasm that lacks the malignant features of the epithelioid component. The poorly differentiated variant of synovial sarcoma is described in the literature as reminiscent of Ewing sarcoma/primitive neuro-ectodermal tumors with small round blue cells or epithelioid cells along with spindle cells of high-grade morphology [109]. Positive staining for TLE1 on cell block sections is seen in synovial sarcomas, while detection of *EWSR1* rearrangement is helpful in diagnosing Ewing sarcoma/primitive neuro-ectodermal tumors. Ewing sarcoma/primitive neuro-ectodermal tumors can have cytomorphology similar to T-LLLL, and both entities express CD99; however, flow cytometry can be helpful to distinguish lymphoma [111].

### 3.3. Germ Cell Tumors (GCTs)

Primordial germ cells retained or misplaced during migration in embryologic development are thought to give rise to extragonadal GCTGCT [112,113]. Extragonadal GCTs predominantly occur in the retroperitoneum; however, mediastinal GCTs can account for 16% of mediastinal masses and have a bimodal age distribution [114,115]. The presentation of mediastinal GCTs at differing ages can cause some difficulty in diagnostic considerations, especially in the absence of a history of a gonadal mass. Although FNA has been shown to be diagnostically accurate in identifying GCTs, clinical history (elevated serum markers of alpha fetoprotein, human chorionic gonadotropin, and lactate dehydrogenase) and radiologic history are crucial in diagnosis GCTs [116]. Rearrangement of isochromosome 12p is characteristic of GCTs and can be detected via FISH on cell block sections [117]. GCTs can be further categorized into seminomas and non-seminomatous GCTs.

#### 3.3.1. Seminoma

The cytomorphology of seminomas of testicular origin and mediastinal origin is identical, which is similar to most GCTs. Therefore, correlation with radiographic imaging is paramount to exclude a gonadal primary. Seminomas represent 3–4% of mediastinal tumors [118,119]. The seemingly dual population of epithelioid cells and lymphocytes can cause diagnostic confusion with thymoma, as discussed previously. The cells of seminoma can be single or in loose clusters, polygonal with enlarged nuclei, and can have prominent nucleoli and, in some cases, vesiculated chromatin (Figure 17). The background lymphocytes are small, and occasional granulomas or giant cells can be seen [120]. The characteristic tigroid background is best visualized with air-dried Diff-Quik smears; however, this phenomenon is notably not entirely specific to seminoma [121,122]. In contrast, the epithelioid cells of thymoma are less polygonal, have a coarser chromatin pattern, and are more intricately associated with background lymphocytes. In addition, large-cell lymphomas can have moderate amounts of cytoplasm and prominent nucleoli, but lymphoglandular bodies should be present. In these cases, flow cytometry and lymphoma markers on cell block preparations can be helpful. The Reed-Sternberg cells of Hodgkin lymphoma can also have moderate cytoplasm and prominent nucleoli, but their background lymphocytic population should be polymorphous rather than small. The cells of seminoma are more polygonal, with clear cytoplasm and a vesicular chromatin pattern. A case of mediastinal seminoma misinterpreted as anaplastic thyroid carcinoma has also been reported [123]. Cell block preparations for immunohistochemical studies are helpful as seminomas will be positive for PLAP, CKIT, OCT3/4, SOX17, and SALL4 and negative for CD30 and AFP [119,123,124,125]. Weissferdt and colleagues demonstrated that mediastinal seminomas can have CAM5.2 expression, albeit focally compared with thymomas and thymic carcinomas, which have diffuse staining. Additionally, MAGEC2 was shown to be sensitive to seminomas [119]. Mediastinal seminomas tend to have a better prognosis than non-seminomatous mediastinal GCTs, excluding mature cystic teratomas [125,126]

#### 3.3.2. Non-Seminomatous GCTs

As teratomas arise from any two or three germ cell layers—endoderm, mesoderm, or ectoderm—a variety of tissue types can be seen on FNA smears, including cartilage, bone, respiratory epithelium, and squamous epithelium [127]. Teratomas can also contain mature or immature somatic elements, depending on the degree of differentiation, and are most commonly present as an anterior mediastinal mass that may show calcifications, bone, or even teeth on CT [128,129]. Cystic changes are often seen in mature teratomas, whereas immature teratomas are often solid masses. Cystic changes can pose diagnostic dilemmas. These changes, along with the presence of thymic tissue and seemingly “normal” tissue, may cause a teratoma diagnosis to be overlooked. Should the teratoma be a component of a mixed GCTGCT, thorough sampling is warranted to reach the correct diagnosis.

With greater pleomorphism of the neoplastic cells compared with seminoma, embryonal carcinomas may have minimal to moderate cytoplasm of epithelioid cells, enlarged irregular nuclei, and prominent nucleoli [130]. The pattern of cells can be glandular, solid, or even papillary with background hemorrhage and necrosis [120]. Immunohistochemical stains show that embryonal carcinomas are positive for SALL4, PLAP, OCT3/4, CKIT, and CD30. Yolk sac tumors occur predominantly in younger male patients and exhibit a wide variety of patterns, such as reticular, solid, endodermal sinus, myxoid, glandular, papillary, hepatoid, and alveolar. Irregular, large, cohesive groups of cells are present on FNA smears. Schiller Duval bodies or hyaline globules can also be present. Given the various morphological patterns, embryonal carcinomas can be confused with metastatic carcinomas such as hepatocellular carcinoma due to the hepatoid pattern, adenocarcinoma due to glandular or papillary patterns, and even thymic carcinomas [120]. A robust immunohistochemical panel in which yolk sac tumors would be positive for CAM5.2, SALL4, AFP, and glypican-3 while negative for OCT3/4, CKIT, and, in general, CD30 is helpful on cell block preparations, as is the correlation with elevated serum AFP levels [131].

Choriocarcinomas can cause diagnostic dilemmas with pleomorphic carcinomas, melanomas, or even mesothelioma. The key to recognition, although choriocarcinoma is a rare entity even among non-seminomatous GCTs, is a dual cell population. Cytotrophoblastic cells have clear cytoplasm with round nuclei and prominent nucleoli, in contrast with syncytiotrophoblastic cells that are multi-nucleated and have pleomorphic nuclei [132,133]. Hemorrhage and necrosis can also be present. The syncytiotrophoblasts often surround the mononuclear cytotrophoblast clusters [120]. Consistently positive for pan-cytokeratin and human chorionic gonadotropin, choriocarcinomas are generally negative for other germ cell markers [132]. A correlation with elevated serum beta- human chorionic gonadotropin levels is certainly necessary.

Mixed GCTs in the mediastinum account for 20% of mediastinal GCTs, in which various tumor components or types of GCTs can be present [134,135]. Somatic and hematologic malignancies can also arise in GCTs. As associated somatic and hematologic malignancies have also demonstrated isochromosome 12p, a common origin for GCTGCT has been postulated [136,137,138]. Hematologic malignancies most commonly occur in association with mediastinal non-seminomatous GCTs [139].

### 3.4. Hematologic Malignancies

Lymphoproliferative disorders of the mediastinum represent approximately 50–60% of mediastinal malignancies in children and adults [140]. More commonly, the mediastinum is involved secondarily in systemic disease; however, in 5–10% of all lymphomas, primary mediastinal involvement does occur [140]. The thymus, mediastinal lymph nodes, and/or extranodal mediastinal organs, in the absence of systemic disease are considered primary involvement. For the reason that a discussion of hematologic malignancies would be vast, we will discuss the more commonly encountered or unique entities here, including classical Hodgkin lymphoma, primary mediastinal large B-cell lymphoma, mediastinal gray zone lymphoma, and T-LL. Our previously reported series on endobronchial ultrasound-guided FNA shows a greater success rate and higher sensitivity and accuracy rates for the diagnosis of lymphomas in mediastinal lymph nodes [141].

#### 3.4.1. Classical Hodgkin Lymphoma (CHL)

Nodular sclerosing CHL (NS CHL) is the most common subtype, accounting for 70% of CHL cases. These lymphomas present with bulky anterior mediastinal disease, with half of the cases arising from the thymus. CHL is more common in younger female patients, whereas mixed cellularity CHL occurs more often in older male patients, involves the middle mediastinum, and is associated with Epstein-Barr virus, with immunocompromised patients being particularly at risk [142,143,144].

NS CHL on FNA smears can show fragments of the characteristic fibrous bands with a polymorphous background, including lymphocytes, macrophages, eosinophils, neutrophils, and plasma cells. Interspersed are large multi-nucleated or binucleated Reed-Sternberg cells with fine chromatin and prominent eosinophilic or basophilic nucleoli. Several variations of the Reed-Sternberg cell can be seen, such as those that are mono-lobated or have glassy eosinophilic cytoplasm [145]. The Reed-Sternberg cells can also form aggregates and can be described as a syncytial variant when conforming into a more sheet-like pattern [145]. Thymic involvement can show remnants of thymic tissue and cystic changes. Mixed cellularity CHL rarely involves the thymus and lacks the fibrous bands of NS CHL. On cell block preparations, immunohistochemical studies can be helpful as the Reed-Sternberg cells are positive for CD30 and MUM1, weakly positive for PAX-5, and positive for CD15 in up to 85% of cases. They are negative for CD45 and usually negative for CD20.

Should the Reed-Sternberg cells cluster or appear in aggregate, they can resemble carcinomas or thymic epithelial neoplasms. In such cases, the epithelial cells of TEN are smaller than those of Reed-Sternberg and are intimately associated with the lymphoid tissue of the thymus. The epithelial cells will be positive for p40 or pan-cytokeratin, whereas Reed-Sternberg cells will be negative, and the background lymphoid tissue in CHL is polymorphous with eosinophils and plasma cells, unlike in TEN cells. The epithelial cells of TEN will show a nested growth pattern and distinct cell membranes, with background thymocytes positive for TdT and CD1a. Carcinomas or melanomas may have greater pleomorphism and larger aggregates, unlike scattered Reed-Sternberg cells. Immunohistochemical stains for pan-cytokeratin for carcinomas or SOX10, S100, and other melanoma markers would be helpful to rule out metastatic melanoma. Anaplastic large cell lymphoma can also be positive for CD30 in its bizarre pleomorphic cells, which can be confused for Reed-Sternberg cells. However, anaplastic large-cell lymphomas are negative for CD15, have some T-cell marker expression in most cases, and are ALK-positive in 50% of cases.

#### 3.4.2. Primary Mediastinal Large B-Cell Lymphoma

Diagnostic dilemmas can occur when distinguishing CHL from other hematologic malignancies, such as primary mediastinal large B-cell lymphoma (PMLBCL). PMLBCL more commonly occurs in young women, arising in the anterior mediastinum from thymic medullary B-cells, and is a distinct aggressive subtype of LBCL, which accounts for 2–3% of all non-Hodgkin lymphomas [146]. Although LBCL disease is localized, the disease can infiltrate adjacent mediastinal structures, leading to compressive symptoms [147]. PMLBCL and secondary involvement of the mediastinum by LBCL show large cells with irregular nuclear contours without the polymorphous lymphoid background seen in CHL. The nuclei may have a vesicular chromatin pattern, and the clear cytoplasm in some cells can lead to the surrounding rim of the cell [147,148]. Sclerosis can be seen on FNA smears, leading to extensive crush artifacts. B-cell markers, such as CD20, strong PAX5, and CD79a, will be positive on immunostaining, along with CD45, which is notably not seen in CHL. MUM1, CD23, and BCL6 can also be positive in PMLBCL. Strong CD30 and CD15 expression in Reed-Sternberg cells would favor CHL, although weak staining can be seen for CD30 and CD15 in LBCLs and PMLBCLs [149]. Unlike LBCL, PMLBCL is negative for rearrangements in BCL6, BCL2, or MYC but can have overexpression of *MALMAL* [147,150]. Increased levels of *REL* at 2p13-16 and *JAK2* at 9p24 can be seen [151].

#### 3.4.3. Mediastinal Gray Zone Lymphoma

Adding to the complications of diagnosing hematologic malignancies, mediastinal gray zone lymphoma shows features intermediate between CHL and LBCL. Molecular and genetic studies have shown CIITA translocations, amplifications of *REL* (2p16.1) and *JAK2* (9p24.1) in all three entities, demonstrating that they are closely related [152,153,154,155]. Mediastinal gray zone lymphomas typically occur in young male patients and present as large anterior mediastinal masses. The morphologic and immunophenotypic features are also intermediate to CHL and LBCL, with a greater quantity and pleomorphism of tumor cells compared with those of CHL and LBCL. Mediastinal gray zone lymphomas have a background inflammatory infiltrate consisting of lymphocytes, plasma cells, and eosinophils [156]. Occasionally, Hodgkin-like or Reed-Sternberg–like cells may be present with strong expression of B-cell markers and CD45 [155,156,157]. Immunophenotypically, mediastinal gray zone lymphomas are positive for CD45 and CD30 and are usually positive for CD15, CD20, PAX5, CD79a, OCT2, and BOB1 [149]. Despite the predominant cytomorphology, MAL positivity may be present. Composite tumors in which both CHL and LBCL are present or in which these lymphomas sequentially present are not considered mediastinal gray zone lymphomas [156].

#### 3.4.4. T-Cell Lymphoblastic Lymphoma

Risk factors for the development of T-LL and acute lymphoblastic lymphoma include syndromes such as ataxia telangiectasia and environmental factors such as exposure to ionizing radiation or lymphotropic viral infections [158,159,160]. Most lymphoblastic lymphomas are of the T-cell immunophenotype and are mainly seen in male patients who are 3–30 years old, with one study finding a median age of 10 years [38]. T-LL presents as a mediastinal mass with minimal to no bone marrow involvement. These young patients can present with chest pain, superior vena cava syndrome, pleural and pericardial effusions, and shortness of breath [161]. FNA smears are generally hypercellular and show discohesive lymphoblasts that range from small to large in size with a high nuclear-to-cytoplasmic ratio and minimal basophilic cytoplasm (Figure 18). The small-to-intermediate-size lymphoblasts have inconspicuous nucleoli, whereas prominent nucleoli can be present in larger lymphoblasts. Irregular nuclear membrane or clefting can be present, and “hand mirror-shaped” cytomorphology has been described in the literature for larger lymphoblasts [38]. Compared with the blastoid variant of mantle cell lymphoma, T-LL does not have irregularly enlarged nucleoli, nor does it have the vacuolated cytoplasm that is classic for Burkitt lymphoma [38]. Additionally, T-LL is more common in younger patients. In such cases, ROSE can be particularly vital in triaging specimens for immunophenotyping by flow cytometry to demonstrate T-cell lineage in contrast to the B-cell lineages of mantle cell lymphoma and Burkitt lymphoma.

As discussed previously, the mediastinal location of T-LL can render diagnostic confusion with lymphocyte-predominant thymomas, although thymomas are typically present in adult patients. Nevertheless, the non-neoplastic immature thymocytes of thymomas can appear indistinguishable from the small T lymphoblasts. However, careful inspection reveals a gradation of maturation that should be present for thymocytes in contrast to the lymphoblasts of T-LL. Recognition of an epithelial component, especially as tissue fragments, is paramount, and immunohistochemical studies for pan-cytokeratin, CK5/6, or p40 on cell block sections can help in this endeavor. The gradation or smear pattern is also reflected in the plots for flow cytometry for surface CD3, CD45, CD4, and CD8 for thymomas or thymic hyperplasia. The thymocytes can be CD4-positive/CD8-negative, CD4-negative/CD8-positive, or can have variable expression of both, resulting in a “double-tailed comet” on flow plot [38,162]. In contrast, T-LL shows a cluster pattern of CD3, CD45, CD4, and CD8 on flow cytometry [33,37]. It is important to note that both entities can show expression of CD1a, cytoplasmic CD3, partial surface CD3, TdT, CD34, CD10, CD45 (dim), and CD99, as well as double expression of CD4/CD8. Aberrant expression of CD10 and CD34 can be seen in T-LL, along with loss of expression of T-cell antigens, which is not commonly seen in thymocytes [37,38,162]. In a previous series, although an immature T-cell phenotype was detected by flow cytometry from an FNA specimen initially diagnosed as T-LL, molecular studies using PCR for T-cell receptor rearrangement were negative. The subsequent resection demonstrated improved histological presence of epithelial cells, confirmed with pan-cytokeratin positivity, rendering a final diagnosis of thymoma. Therefore, the interpretation of immature T-cell phenotypes on flow cytometry of a mediastinal mass should be approached with caution, and particular attention should be paid to flow plot patterns.

## 4. Conclusions

Image-guided FNA can be readily used as a minimally invasive procedure in sampling and specimen triage. Utilization of ROSE also increases the efficacy of specimen triage for optimum patient care.

## Figures and Tables

**Figure 1 diagnostics-13-02400-f001:**
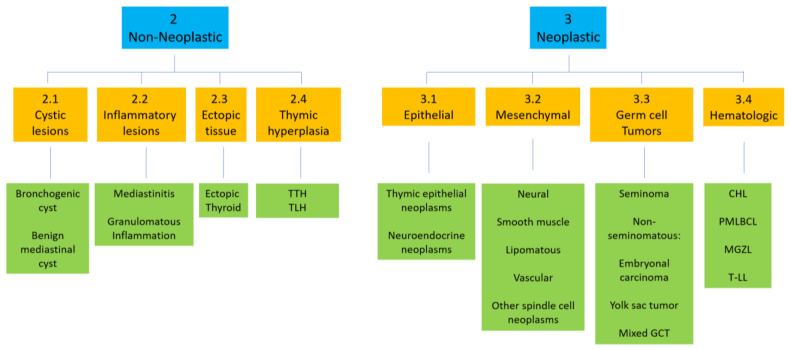
General schema and examples for the cytologic diagnoses on fine needle aspiration of mediastinal lesions.

**Figure 2 diagnostics-13-02400-f002:**
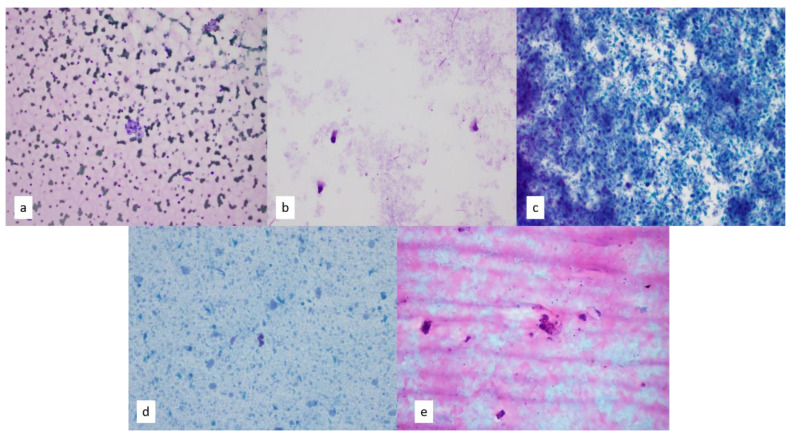
(**a**) Foamy macrophages within a background of proteinaceous fluid are indicative of cystic changes on Diff-Quik smear, 4×. (**b**–**d**) Papanicolaou-stained smears on Diff-Quik smear (**b**) and at 10× magnification (**c**,**d**) show degenerated cells, amorphous debris and ciliated epithelium in a bronchogenic cyst. (**e**) Papanicolaou-stained smear shows proteinaceous fluid and rare epithelial groups, indicating a diagnosis of cystic thymoma on fine needle aspiration, 10×. Concurrent core needle biopsy of the same lesion confirmed multilocular thymic cyst.

**Figure 3 diagnostics-13-02400-f003:**
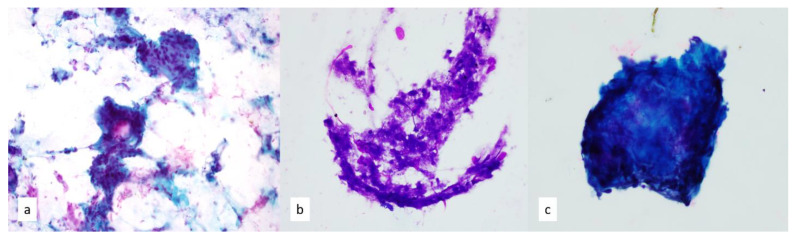
(**a**) Papanicolaou-stained smear, 20×. Epithelioid to spindled histiocytes cluster in aggregates in granulomatous inflammation. (**b**) Diff-Quik-stained smear of necrotizing granulomas show focal areas of necrosis, 20×. (**c**) Papanicolaou-stained smear, 40×. FNA of sclerosing mediastinitis can be cytologically challenging with only scant fragments of sclerotic tissue.

**Figure 4 diagnostics-13-02400-f004:**
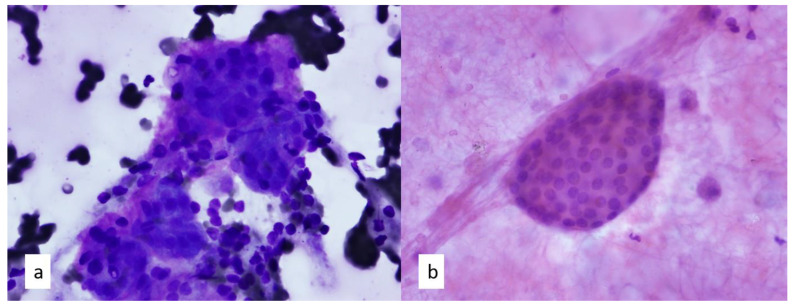
(**a**) Diff-Quik–stained-smear and (**b**) Papanicolaou-stained smear of thyroid follicular epithelial cells with round nuclear contours and coarse chromatin pattern are arranged in tight packets in a background of colloid on FNA of ectopic thyroid tissue, 40×.

**Figure 5 diagnostics-13-02400-f005:**
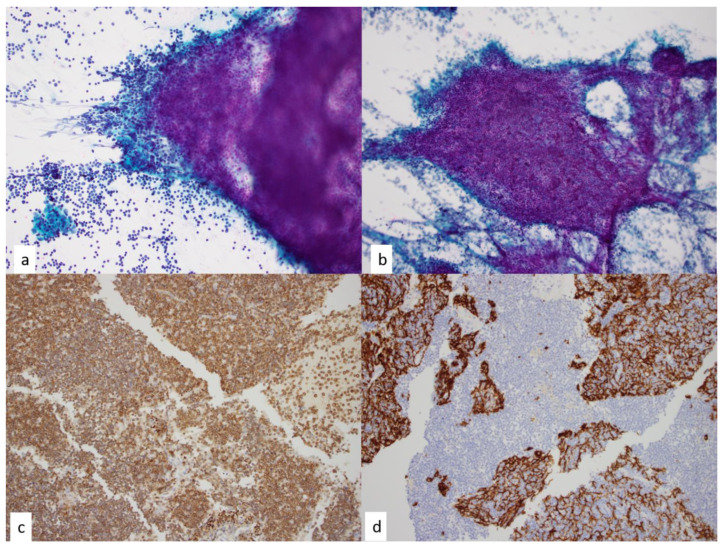
Thymic epithelial neoplasms are composed of neoplastic epithelial cells and background thymocytes. (**a**) In lymphocyte predominant cases, the epithelial component may be inconspicuous. This Papanicolaou-stained smear shows a rare epithelial fragment (bottom left) in a lymphocyte predominant thymoma, 10×. (**b**) Papanicolaou-stained smear of lymphocyte-predominant thymoma in which FNA smears demonstrate clusters of thymocytes and the rare neoplastic epithelial cells are interspersed, 4×. (**c**) An immunohistochemical stain performed on a cell block section highlights the thymocytes as positive for CD3, 4×. (**d**) Cytokeratin cocktail immunohistochemical analysis demonstrates neoplastic epithelial cells, 4×.

**Figure 6 diagnostics-13-02400-f006:**
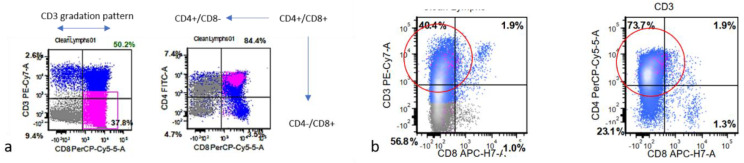
(**a**) Immunophenotyping by flow cytometry shows a normal gradation pattern of thymocytes in varying maturation stages as CD4 negative/CD8 positive, CD4 positive/CD8 positive, and CD4 positive/CD8 negative. (**b**) In contrast, T-lymphoblastic leukemia/lymphoma shows predominantly tight clustering of CD3, and in this example, CD4 positive/CD8 negative cells without CD4 positive/CD8 positive thymocytes (circled area).

**Figure 7 diagnostics-13-02400-f007:**
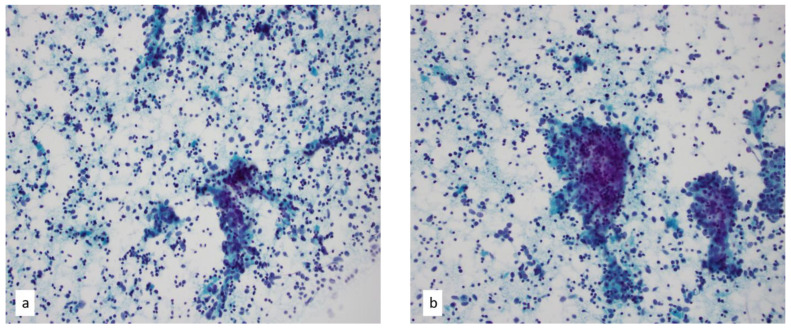
(**a**,**b**) Papanicolaou-stained smear of thymoma with equal components of the neoplastic epithelial cells and background thymocytes of T-cell origin on a fine needle aspiration, 40×. Loose clusters of epithelial with round to oval nuclei in a background of small, mostly mature-appearing lymphocytes are seen.

**Figure 8 diagnostics-13-02400-f008:**
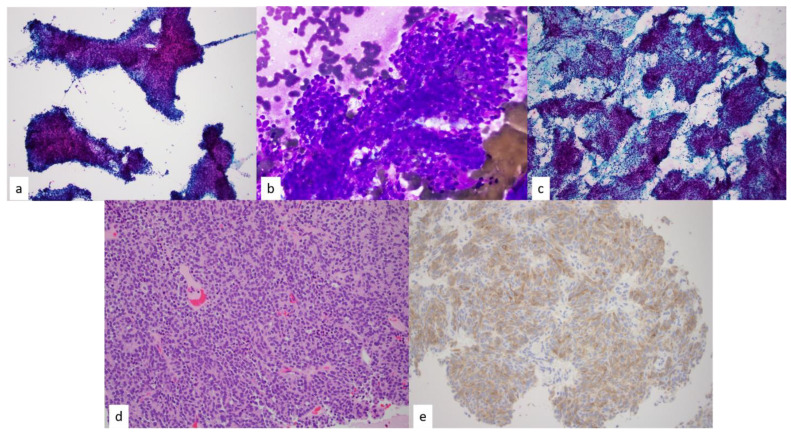
(**a**) Papanicolaou-stained smear of epithelial-predominant thymomas demonstrate fragments of neoplastic epithelial cells with scant to minimal lymphocytes, 4×. Circular spaces within the fragments are commensurate to microcystic changes commonly seen on histology of thymomas. Diff-Quik (**b**) Diff-Quik-stained smear of carcinoid tumor arises within the differential diagnosis of epithelial predominant thymomas, 40×. (**c**) Papanicolaou-stained smear, 4×. In this case, the fine needle aspiration cytology diagnosis and corresponding core needle biopsy were diagnosed as well-differentiated neuroendocrine carcinoma (Carcinoid tumor). (**d**,**e**) However, final resection confirmed (**d**) a spindle cell thymoma with neuroendocrine differentiation hematoxylin and eosin staining, 20×, and (**e**) positive chromogranin on immunohistochemistry.

**Figure 9 diagnostics-13-02400-f009:**
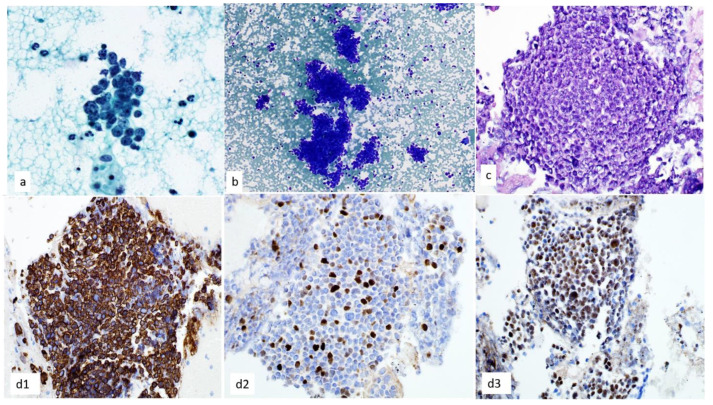
(**a**,**b**) NUT carcinoma shows loosely cohesive clusters with small to medium-sized relatively monotonous cells, high nuclear/cytoplasmic ratio, and variable chromatin with central nucleolus on (**a**) Papanicolaou-stained smear, 60×, and (**b**) Diff-Quik-stained smear, 20×. (**c**) A cell block section shows tumor cells on hematoxylin and eosin staining, 40× (**d**) Immunohistochemical stains are positive for (**d1**). CK5/6, (**d2**). p40, and (**d3**). NUT, 40×.

**Figure 10 diagnostics-13-02400-f010:**
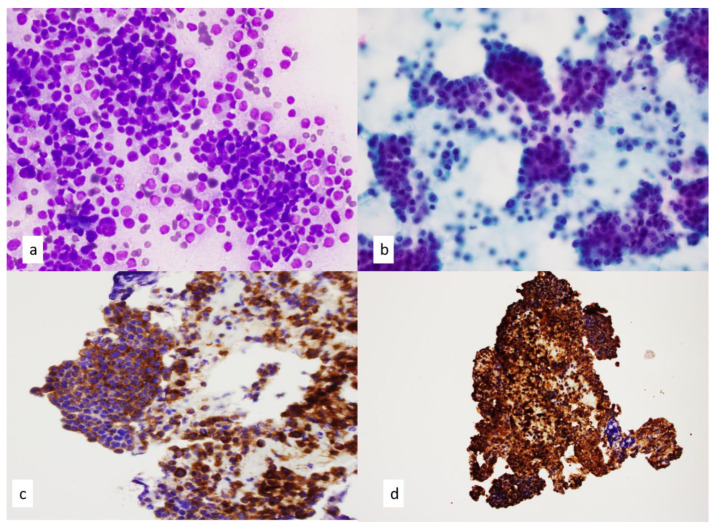
(**a**,**b**) FNA smears of well-differentiated neuroendocrine carcinoma demonstrate an organoid growth pattern of uniform round to ovoid cells. Single cells with scant cytoplasm or even stripped nuclei show a finely stippled or a salt-and-pepper chromatin pattern on (**a**) Diff-Quik staining, 60×, and (**b**) Papanicolaou staining, 60×. (**c**,**d**) Immunohistochemical stains performed on cell block sections are positive for (**c**) synaptophysin, 20× and (**d**) chromogranin 10×.

**Figure 11 diagnostics-13-02400-f011:**
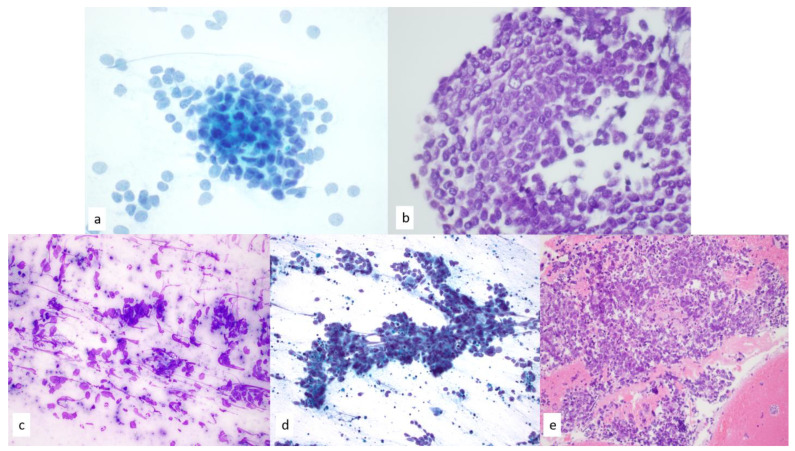
(**a**,**b**) Cell block section of moderately differentiated neuroendocrine carcinoma shows round to ovoid cells with minimal cytoplasm and finely stippled chromatin pattern on (**a**) Papanicolaou staining, 60×, and (**b**) hematoxylin and eosin staining, 60×. Focal area of apoptosis and a rare mitotic figure are present; however, no areas of necrosis were identified. (**c**,**d**) Small cell carcinoma demonstrates (**c**) “smudge artifact” on Diff-Quik–stained smears, 60× and (**d**) numerous apoptotic bodies on Papanicolaou-stained smears, 40×. (**e**) Nuclear molding and areas of necrosis are present on hematoxyin and eosin–stained cell block sections, 20×.

**Figure 12 diagnostics-13-02400-f012:**
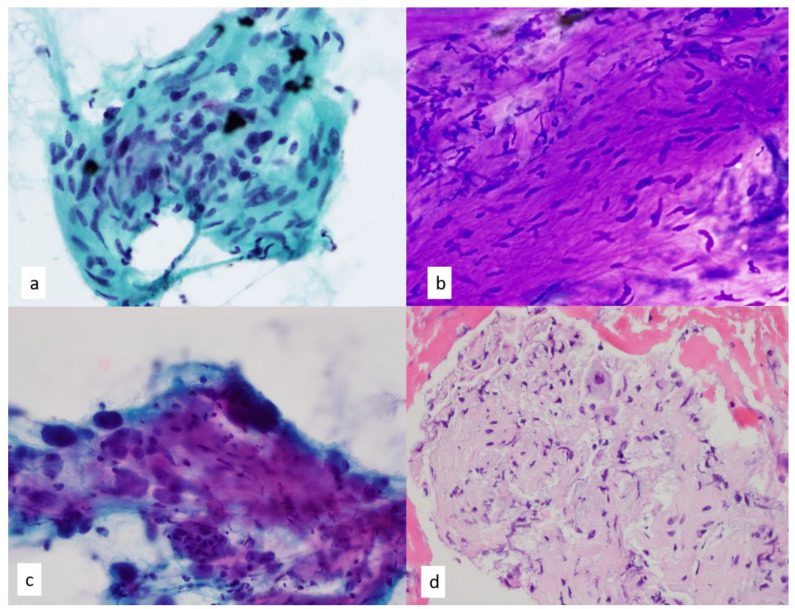
(**a**) Fine needle aspiration smears show narrow, elongated, and tapered nuclei within a fibrillary stroma, and a rare intranuclear inclusion is present in a case of schwannoma of the posterior mediastinum, Papanicolaou-stained smear, 60×. (**b**) Schwannian cells on Diff-Quik staining of ganglioneuroma demonstrate intersecting fascicles of spindle cells with wavy tapered nuclei in a myxoid-like stroma, 60×. (**c**) Numerous ganglion cells are seen with abundant granular cytoplasm and round eccentric nuclei with prominent nucleoli. Binucleated forms are also present. Papanicolaou-stained smear, 60× (**d**) Hematoxylin and eosin stain of a cell block section of ganglioneuroma shows a spindle cell proliferation of Schwann cells and rare ganglion cell, 40×.

**Figure 13 diagnostics-13-02400-f013:**
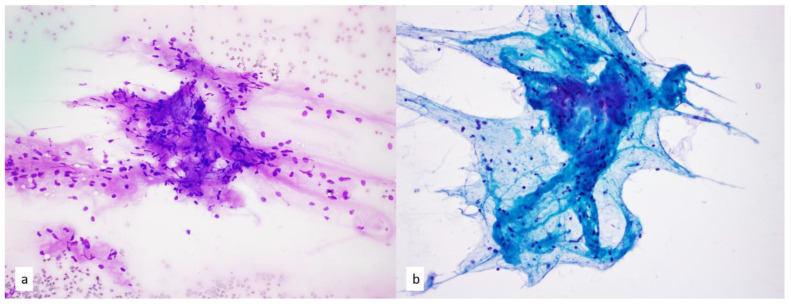
(**a**) Diff-Quik and (**b**) Papanicolaou-stained smears of neurofibroma, 40×. Scant fragments of spindle cells with indistinct cytoplasm and wavy nuclei with tapered ends are present. FNA is often scant or non-diagnostic in cases of neurofibroma.

**Figure 14 diagnostics-13-02400-f014:**
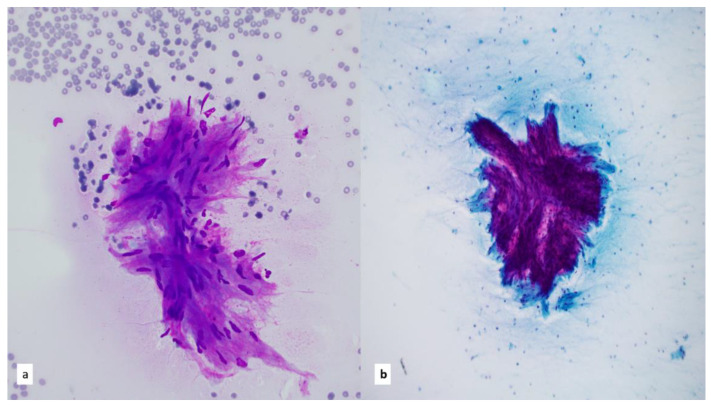
Fine needle aspiration shows spindle cell proliferation arranged in fascicles, elongated nuclei with blunted ends, and eosinophilic cytoplasm. No necrosis or mitotic activity was identified. Corresponding core needle biopsy confirmed smooth muscle tumor of undetermined malignant potential, mainly due to the small sample size, (**a**) Diff-Quick smear, 60×; (**b**) Papanicolaou-stained smear, 40×.

**Figure 15 diagnostics-13-02400-f015:**
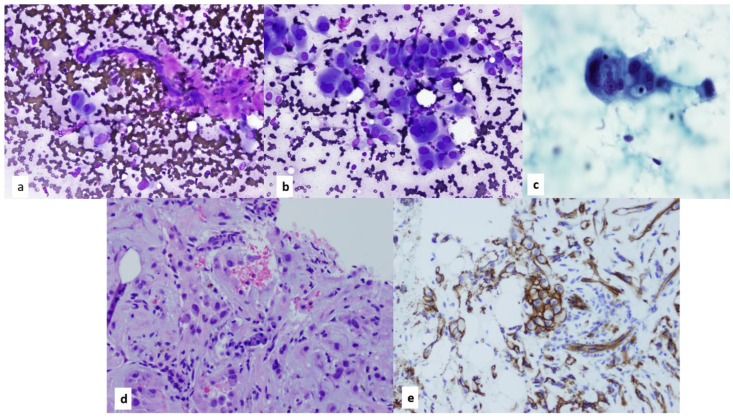
(**a**) Single polygonal cells and adjacent metachromatic myxofibrillary stroma surrounding a vascular structure in a Diff-Quik smear of epithelioid hemangioendothelioma, 20×. (**b**) The polymorphic cells of EHE are in loose clusters, some adopting single file cord arrangement, Diff-Quik stain 40× (**c**,**d**) Papanicolaou stained smear (60×) and cell block section (40×) show tumor cells with enlarged ovoid nuclei and coarse chromatin pattern, prominent nucleoli, intranuclear inclusions and cytoplasmic vacuoles that contain red blood cells. (**e**) CD34 immunohistochemistry (40×) on cell block sections demonstrate the vascular nature of EHE.

**Figure 16 diagnostics-13-02400-f016:**
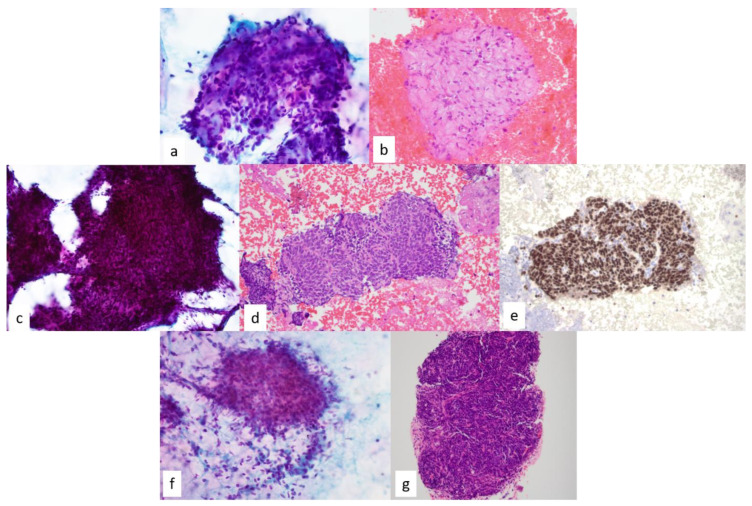
(**a**,**b**) Papanicolau -stained smear (60×) and hematoxylin and eosin-stained cell block section (20×) demonstrate clusters of ovoid to spindled cells with minimal cytoplasm and few scattered isolated cells. The nuclei are hyperchromatic with no intranuclear inclusions identified and embedded within a collagenous stroma with irregular edges. Final cytologic diagnosis was solitary fibrous tumor (**c**) In contrast, monophasic synovial sarcoma shows tight groups of spindled to ovoid cells with scant cytoplasm and smooth elongated nuclei. Ropy bands of collagenous stroma seen in SFT are not present however transgressing vessels, although not specific, can be seen, Papanicolaou -stained smear, 40×; hematoxylin and eosin–stained cell, 20×. (**d**) Immunostain for TLE-1 performed on a cell block section shows strongly positive nuclear staining, in keeping with synovial sarcoma, 20×. (**e**,**f**) Spindle cell thymoma causes diagnostic dilemmas with other spindle cell neoplasms. (**g**) Recognition of background lymphocytes and utilizing keratin stains to delineate the epithelial nature of the neoplasm is helpful, Papanicolaou -stained smear, 40×; hematoxylin and eosin–stained cell, 10×.

**Figure 17 diagnostics-13-02400-f017:**
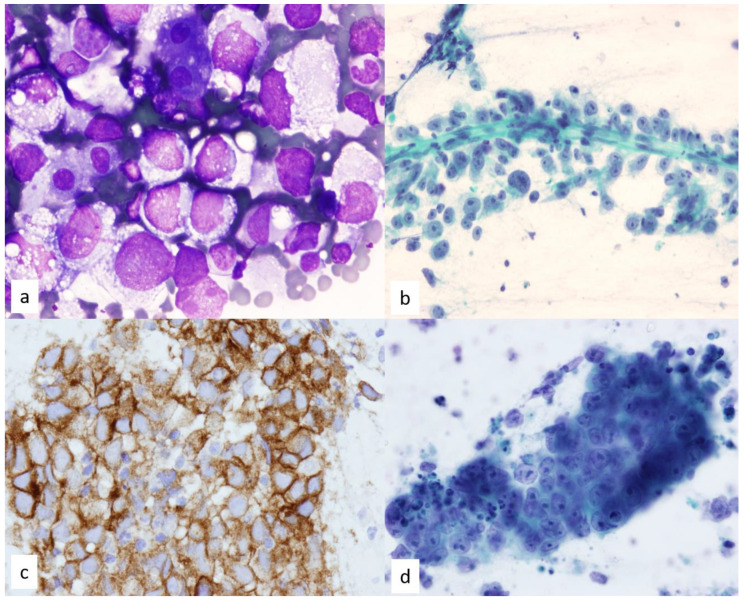
(**a**) Diff-Quik staining of a case of seminoma shows poorly cohesive cells with hyperchromatic enlarged nuclei and moderate amount of clear to vacuolated cytoplasm, 60×. Few background lymphocytes are present. (**b**) Prominent nucleoli in poorly cohesive cells surround vascular structures in this Papanicolaou-stained FNA smear of seminoma, 40×. (**c**) Immunohistochemistry performed on a cell block section for seminoma is positive for C-Kit, 20×. (**d**) Large polygonal cells with round or oval nuclei, distinct nucleoli, and pale staining of the cytoplasm with indistinct borders are seen in embryonal carcinoma. The tumor cells tend to form groups and clusters, sometimes in a papillary configuration. The pleomorphic nuclei are larger than seminoma with prominent nucleoli.

**Figure 18 diagnostics-13-02400-f018:**
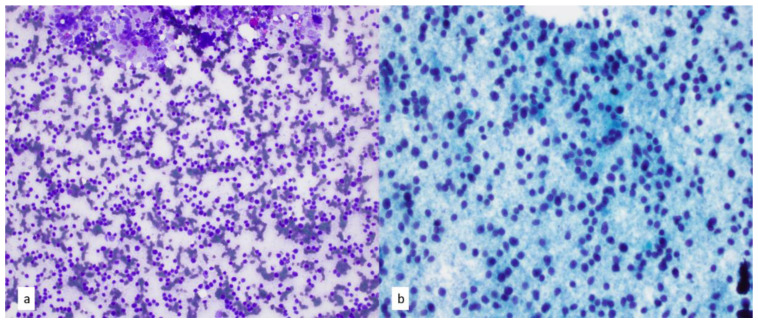
(**a**) FNA smears of T-lymphoblastic leukemia/lymphoma show lymphocytes with minimal cytoplasm and delicate chromatin pattern. Intermixed macrophages have a light-to-dark pattern on Diff-Quik stain, 20×. (**b**) Papanicolaou-stained smear of T-cell lymphoblasts shows occasional nuclear grooves and notching, 40×.

**Table 1 diagnostics-13-02400-t001:** List of Abbreviations.

ANNUBP	Atypical neurofibroma of uncertain malignant potential
CHL	Classical Hodgkin lymphoma
EBUS	Endobronchial ultrasound
EUS	Endoscopic ultrasound
FNA	Fine needle aspiration
GCT	Germ cell tumor
HM	Hematologic malignancy
IHC	Immunohistochemistry
LBCL	Large B-cell lymphoma
MC CHL	Mixed cellularity classical Hodgkin lymphoma
MDNEC	Moderately differentiated neuroendocrine carcinoma
MGZL	Mediastinal gray zone lymphoma
MN	Mesenchymal neoplasm
MPNST	Malignant peripheral nerve sheath tumor
ND	Non-diagnostic
NE	Neuroendocrine neoplasm
NF	Neurofibromatosis
NS CHL	Nodular sclerosing classical Hodgkin lymphoma
PDNEC	Poorly differentiated neuroendocrine carcinoma
PMLBCL	Primary mediastinal large B-cell lymphoma
ROSE	Rapid on-site evaluation
SFT	Solitary fibrous tumor
TEN	Thymic epithelial neoplasms
T-LL	T-lymphoblastic lymphoma
WDNEC	Well differentiated neuroendocrine carcinoma

## Data Availability

Data sharing is not applicable to this article as no new data were created or analyzed in this study.

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
