# Peer review of "The Utility of Fine Needle Aspiration (FNA) Biopsy in the Diagnosis of Mediastinal Lesions"

_diagnostics, 2023, doi:10.3390/diagnostics13142400_

Round 1

Reviewer 1 Report

Jun 15, 2023

Dear Author,

 1.     The manuscript, diagnostics-2474831, is within the scope of the journal.

2.     The section “Abstract” should be revised.

3.     In my opinion, minimal invasive FNA with smaller needle sizes and ThyMIFNAis an important issue. Therefore we have also emphasized its cruciality, to date. So, we evaluate the author’s hypothesis as valuable. To this end, the section “Discussion” must be revised, enriched by discussing this essential issue, and the reference list must be enriched with several current and updated articles, including “Proposal of a novel terminology: Minimally invasive FNA and Thyroid minimally invasive FNA; MIFNA and Thyroid MIFNA.”, “Thyroid minimally invasive FNA (Thy MIFNA); Proposal of novelty in terminology.”, and “Minimum minimorum: Thy MIFNA, less is more concept? Volens nolens?”

4.     The orthographical and grammatical errors, should be revised by the authors.

5.     The manuscript might be accepted for publication in Diagnostics after minor revision.

Best Regards,

Reviewer, Diagnostics

Minor editing of English language required.

Author Response

We would like to thank the reviewer for their time and consideration for reviewing our manuscript and are appreciative of the comments and suggestions. Please see below for our responses.

  1. The manuscript, diagnostics-2474831, is within the scope of the journal.
  2. The section “Abstract” should be revised.

Response: Please let us know what should be revised within the abstract. However, we believe the abstract highlights all the main points of the text as needed.

  1. In my opinion, minimal invasive FNA with smaller needle sizes and ThyMIFNAis an important issue. Therefore we have also emphasized its cruciality, to date. So, we evaluate the author’s hypothesis as valuable. To this end, the section “Discussion” must be revised, enriched by discussing this essential issue, and the reference list must be enriched with several current and updated articles, including “Proposal of a novel terminology: Minimally invasive FNA and Thyroid minimally invasive FNA; MIFNA and Thyroid MIFNA.”, “Thyroid minimally invasive FNA (Thy MIFNA); Proposal of novelty in terminology.”, and “Minimum minimorum: Thy MIFNA, less is more concept? Volens nolens?”

       Response:  We have reviewed the proposed current publications “Big gain, no pain:Thyroid minimally invasive FNA (Thy MIFNA): Proposal of novelty in terminology” and “Minimum minimorum: thyroid minimally invasive FNA, less is more concept? Volens nolens?” These publications, however, are editorials and their practices are limited to a few places and are not nationally recognized. Fine needle aspiration since its inception was recognized as a minimally invasive procedure and this is one of the reasons that contributed to its popularity and its use for all organ sites. Furthermore, for the scope of this review which focuses on cytomorphology, there is no division of FNAs as to minimally or maximally invasive. The term “minimally invasive” has been incorporated within the introduction and conclusion to emphasize this issue (please see highlighted portion within manuscript).

  1. The orthographical and grammatical errors, should be revised by the authors.

               Response: The manuscript was reviewed and edited by MD Anderson Cancer Center’s institutional editorial office prior to submission.

  1. The manuscript might be accepted for publication in Diagnosticsafter minor revision.

Thank you and kind regards,

Uma Kundu, MD

Reviewer 2 Report

In my opinion, the analyzed topic is interesting enough to attract the readers’ attention. I think that the abstract of this article is well organized and clear.

In my opinion, the discussion could be studied in depth and extended. I think that should be important to evaluate the status of these patients in order to reduce possible postoperative complications. In particular I suggest this article as an example:   The role of preoperative frailty assessment in patients affected by gynecological cancer: a narrative review Ottavia D’Oria, Tullio Golia D’Auge, Ermelinda Baiocco, Cristina Vincenzoni, Emanuela Mancini, Valentina Bruno, Benito Chiofalo, Rosanna Mancari, Riccardo Vizza, Giuseppe Cutillo, Andrea Giannini

Vol. 34 (No. 2) 2022 June, 76-83 doi: 10.36129/jog.2022.34. Because of these reasons, the article should be revised and completed. Figures and tables are interesting.  Considered all these points, I think it could be of interest for the readers and, in my opinion, it deserves the priority to be published after revisions.

A moderate review of English language should be performed.

Author Response

We would like to thank the reviewer for their time and consideration for reviewing our manuscript and are appreciative of the comments and suggestions. Please see below for our responses.

In my opinion, the analyzed topic is interesting enough to attract the readers’ attention. I think that the abstract of this article is well organized and clear.

In my opinion, the discussion could be studied in depth and extended. I think that should be important to evaluate the status of these patients in order to reduce possible postoperative complications. In particular I suggest this article as an example:   The role of preoperative frailty assessment in patients affected by gynecological cancer: a narrative review Ottavia D’Oria, Tullio Golia D’Auge, Ermelinda Baiocco, Cristina Vincenzoni, Emanuela Mancini, Valentina Bruno, Benito Chiofalo, Rosanna Mancari, Riccardo Vizza, Giuseppe Cutillo, Andrea Giannini Vol. 34 (No. 2) 2022 June, 76-83 doi: 10.36129/jog.2022.34. Because of these reasons, the article should be revised and completed. Figures and tables are interesting.  Considered all these points, I think it could be of interest for the readers and, in my opinion, it deserves the priority to be published after revisions.

Response: We agree with the reviewer that minimization of post-operative and post-biopsy complications is exceedingly important for the care of patients. However, usually this role is designated to the proceduralist not the pathologist in radiographically guided FNAs. This review article is mainly to focus on the interpretation of FNAs for appropriate diagnoses.

A moderate review of English language should be performed.

Response: The manuscript was reviewed and edited by MD Anderson Cancer Center’s institutional editorial office prior to submission.

Thank you and kind regards,

Uma Kundu, MD

Round 2

Reviewer 2 Report

I think the manuscript has improved with the changes made and, in my opinion, it deserves priority to be published.